# Erectile dysfunction and its associated factors among the male population in Adigrat Town, Tigrai Region, Ethiopia: A cross-sectional study

**Haftom Tesfay Gebremedhin**[1]*, **Hagos Mehari Mezgebo**[1], **Gessessew Teklebrhan Geberhiwot**[1], **Tesfay Tsegay Gebru**[2], **Yowhans Ashebir Tesfamichael**[2], **Hailu Belay Ygzaw**[1], **Mulu Ftwi Baraki**[3], **Guesh Teklu woledemariam**[2], **Tsegu Hailu Gebru**[2], **Haileslassie Tesfay Tadese**[2], **Gebreslassie Gebreegziabhier Kindeya**[4], **Telake Azale**[5]

1 Department of Psychiatry, College of Medicine and Health Science, Adigrat University, Adigrat, Ethiopia,
2 Department of Nursing, College of Medicine and Health Science, Adigrat University, Adigrat, Ethiopia,
3 Department of Midwifery, College of Medicine and Health Science, Adigrat University, Adigrat, Ethiopia,
4 Department of Psychiatry, College of Health Science, Aksum University, Aksum, Ethiopia, 5 Department of Psychiatry, College of Medicine and Health Science, University of Gondar, Gondar, Ethiopia

* haftomtesfay30@gmail.com

**Data Availability Statement:** All relevant data are within the manuscript and its Supporting Information.

## Abstract

### Background

Erectile dysfunction is one of the common sexual dysfunctions, but it is generally misunderstood as it is not a condition that threatens life. It affects an individual's physical as well as psychosocial health and has a significant impact on sufferers and their families' quality of life. No data are suggesting the prevalence of erectile dysfunction at the population level in Ethiopia. This research aimed to assess the prevalence and associated factors of erectile dysfunction among the male population.

### Methods

We employed a community based cross-sectional study among 802 study participants. A two-stage random sampling method was used for enrolling study participants. Including the International Index of Erectile Function Questionnaire-5 (IIEF-5) for erectile dysfunction, data were collected using pretested and a structured questionnaire administered by an interviewer. Binary logistic regression was performed to identify factors associated with erectile dysfunction.

### Result

Out of the total of 802 individuals, 25.4%(95% CI:(22.4, 28.3%)) (n = 204) reported erectile dysfunction. The mean age of the participants was 34.3 ± 9.6 years. Age of 40years and above [AOR = 10.74, 95% CI: (7.07, 16.35)], physical inactivity [AOR = 3.62, 95% CI: (2.40, 5.45)], depression [AOR = 4.01, 95% CI: (2.22, 7.21)], poor quality of life [AOR = 1.59, 95% CI: (1.07, 2.36)] were significantly associated with erectile dysfunction.

**Funding:** The author(s) received no specific funding for this work.

**Competing interests:** The authors have declared that no competing interests exist.

## Conclusions

In this study, the prevalence of erectile dysfunction was high. Therefore, it is recommended that erectile dysfunction treatment be integrated into the health care system that focuses on educating and inspiring people on healthy eating, physical activity, and behavior enhancing wellbeing.

## Introduction

Erectile dysfunction (ED)has historically been known as impotence [1]. The National Institutes of Health Consensus Conference defined erectile dysfunction as the "consistent inability to achieve or maintain a penile erection, or both, sufficient for an adequate sexual relationship" [2]. In 1995 more than 150 million men were affected by erectile dysfunction worldwide, which is expected to rise to 320 million by 2025 [3]. It is a common medical problem that affects about 15 percent of men annually [4].

Erectile dysfunction impacts physical and psychosocial wellbeing and has a major effect on the quality of life (QoL) of sufferers and their partners and families [5]. Worldwide estimates of the prevalence of erectile dysfunction rates range from 2 percent among men younger than 40 years to 86% among men 80 years of age or older [6]. The Massachusetts Male Aging Study (MMAS), among others, recorded an overall prevalence of 52% erectile dysfunction in non-institutionalized men in the Boston area aged 40–70 years [7]. Whereas in men aged 30 to 80 years in the urban district Cologne study, the prevalence of erectile dysfunction was 19.2% ranging from 2.3% to 53.4% [8]. Moreover, an age-adjusted study conducted in four countries showed a varying prevalence of erectile dysfunction ranging from 15% to 34% [9]. Furthermore, estimates of 18.4% in USA [10], 46.3% in New York state [11], 18.9% in Spain [12], 46.2% Brazil [13], 25.1% in Western Australia [14], 28.3% in China [15], 66.9% in Vietnam [16], 69.2% in Turkey [17], 18.8% in Iran [18], 58.9%in Nigeria [19],24% in Tanzania [20]and 23.5% in Egypt [21]were reported.

Erectile dysfunction may be presented as a result of physical or emotional problems, or a combination of both [22]. Different studies showed potential predictors of erectile dysfunction such as less education, older age, lower socioeconomic status, urban, unemployment, diabetes mellitus, Hypertension, heart disease, lower urinary tract symptoms, liver disease, neurological disorder, arthritis, peptic ulcer disease, prostate problem, depression, physical inactivity, heavy smoking, and alcohol consumption [9, 12, 13, 15–17, 19–21, 23–31].

Although erectile dysfunction is underestimated because it doesn't pose a threat to an individual's life, it is currently one of the most common sexual dysfunctions today [32]. No published data is showing the prevalence of community-level erectile dysfunction in Ethiopia. Therefore, this study was carried out to assess the prevalence and associated factors of erectile dysfunction among the male populations to serve as an input in identification and intervention of male erectile dysfunction by concerned bodies, and as baseline data and recommendations to point out areas for further research by future researchers. It will also offer insight into the program plan.

## Materials and methods

### Study design, setting, and period

A community-based cross-sectional study was conducted in Adigrat Town, Tigrai, Northern Ethiopia, from May 30 to June 30, 2019. According to the Federal Democratic Republic of Ethiopia Central Statistical Agency report, the total population estimated for the Town was 95,358, of which 45,003(47.2%) were male in all age groups [33].

## Study participants

The study included a sample of 845 male individuals aged 18 years and above who had been engaged in sexual activity in the preceding month of the data collection and those who had given written informed consent. Those critically ill and unable to communicate during the data collection period were excluded.

## Sample size and sampling technique

The single proportional formula was used to calculate the sample size, taking into account the following assumptions: proportion with erectile dysfunction 50% since national prevalence is lacking, confidence interval of 95%, 5% margin of error, non-response rate of 10%, and a design effect of 2.0. Accordingly, the sample size was 845.

A two-stage random sampling technique was performed to enroll in study participants. 3 Kebeles were selected by lottery method out of the total six administrative Kebeles in Adigrat town. The overall sample size was assigned to the total number of HH in the selected Kebeles using proportional allocation. A systematic random sampling technique was applied for the selection of study units. The sampling interval or constant number k was nine. It was obtained by dividing total HH in each selected Kebeles to allocated samples of each of the selected Kebeles. The first HH was chosen by the lottery method as a starting point. Then the subsequent households were selected by taking consecutive k[th] households until the allotted sample size for each selected Kebeles was obtained. When more than one eligible study participant was identified in one HH, one individual was selected by the lottery method.

## Data collection

Data were collected using five parts, structured questionnaire with five elements. The first part includes the socio-demographic characteristics of the participants. The second part of the tool is the International Index of Erectile Function (IIEF-5) [34]. The third part of the questionnaire involves the lifestyle of the study subjects, including a history of substance use (such as alcohol use, khat chewing, cigarette smoking) and physical activity. The fourth part of the questionnaire covers the clinical factors of the study participants. The final part of the questionnaire was the World Health Organization Quality of Life Scale-Brief version (WHOQoL-- BREF) intended to assess the quality of life [35]. The WHO QoL-100 scale used in this study is a short version, consisting of 26 items of a self-administered generic questionnaire [35, 36]. This tool is a sound and cross-culturally valid QoL assessment [37].

The data were collected by six Msc psychiatry and supervised by one supervisor after two daysof training on the study tool's administration. During the actual data collection, the supervisor cross-checks the data collection process on randomly selected study units every day. The principal investigator was responsible for coordinating and overseeing the overall data collection process. Tigrigna version questionnaire was used to collect the data.

## Operational definition

**Erectile Dysfunction (ED).** Individuals who scored 1–21 out of 25 points were reported as having ED. In contrast, those who scored 22–25 out of 25 points were reported as not having ED. Those who scored 1–7, 8–11, and 12–21 out of 25 points were classified as severe ED, Moderate ED, and Mild ED, respectively [34].

**Depression.** Those who scored ≥10 out of 27 points from patient health questionnaire-9 (PHQ-9) for depression were considered as having depression. At the same time, those who scored 1–9 out of 27 points from PHQ-9 were considered as having no depression [38].

## Data quality assurance

The questionnaire was initially written in English and translated into Tigrigna's local language by a panel of experts fluent in the language. It was then back-translated into English by an independent person to checkthe consistency of the tool. The Tigrigna version questionnaires were used to collect the data.

A pretest was done two weeks before the actual data collection among 5% (43) sample population at Edaga Hamus, and appropriate modification was made. The tool's reliability was checked using Cronbach's alpha reliability test with a score of 0.897 (95% CI 0.889–0.908). The data were collected by six psychiatric nurses and supervised by one supervisorfollowing two-day training on the administration of the study instruments.

Throughout the data collection process, data collectors were supervised at each site, and regular meetings were held between the data collector, supervisor, and principal investigator to address problematic issues raised by the study participants. The collected data were reviewed and cross-checked for completeness before data entry, and incomplete data were discarded. Data entry format templates were produced and programmed.

## Data analysis

Data were checked, coded, and entered into Epi-Data version 4.2 and exported to SPSS (Statistical Package for Social Science) version 25 for analysis. For describing the socio-demographic characteristics of the study participants, descriptive statistics such as percentage, mean, and standard deviation were used. Tables and figures were also used for the data presentations. Binary logistic regression was used to identify factors associated with erectile dysfunction. Analysis of bivariate logistic regression was carried out to find the association of each independent variable with the outcome variables. To control the effect of potential confounders, all variables with a $p$-value of $\leq 0.2$ at bivariate logistic regression analysis were transferred into the multivariable binary logistic regression model. Variables with a $p$-value of less than 0.05 were deemed statistically significant, and their adjusted odds ratios (AOR) with 95% confidence interval (CI) were calculated. Hosmer and Lemeshow goodness was used to check model fitness (P = 0.199). Multi-collinearity was checked by variance inflation factors (VIF).

## Ethical consideration

Ethical clearance and approval were achieved from the Ethical Review Board of the College of medicine and health sciences, Adigrat University. An official letter of cooperation was received from Tigrai Regional Health Bureauto the administrative office of Adigrat Woreda. The formal letter was then received from Adigrat Woreda's administrative office to each selected kebeles. The study's intent, procedure, benefits, and potential risks had been explained before each respondent's written informed consent was obtained. Unwilling participants were not forced to take part in the study. Interviews were conducted privately to assure confidentiality, and the names of interviewees were not written on the questionnaire. The confidentiality of the information collected was kept throughout the study.

# Result

## Socio-demographic characteristics of the respondents

Out of 845 study participants, 802 individuals participated in the study, giving a response rate of 94.9%. The mean age of the participants was 34.3 ± 9.6 years. More than half (50.7%) of the respondents were married. Almost all (97.5%) of the participants were ethnically Tigrian, whereas the vast majority (90.3%) were Orthodox by religion. Nearly half (49.1%) of the

participants completed college and higher education. About one-third (36%) of the participants were government employees. The majority (79.6%) of the respondents had a monthly income above the poverty line (Table 1).

## Distribution of clinical and lifestyle factors

Regarding the respondents' clinical characteristics, the most frequently occurring chronic medical condition in this study was Hypertension (6%) followed by Diabetes mellitus (5.1%), and about 9.7% of the study participants have depression. Participants' lifestyle revealed that

**Table 1. The socio-demographic characteristic of the respondents of Adigrat Town, North Ethiopia, 2019 (n = 802).**

| Variables | Category | Frequency or percentage |
|---|---|---|
| Age | 18–29 | 335 (41.8%) |
| | 30–39 | 312 (38.9%) |
| | 40–49 | 95 (12.2%) |
| | ≥50 | 57 (7.1%) |
| Marital status | Married | 407 (53.7%) |
| | Single | 345 (40.0%) |
| | Others* | 50 (6.3%) |
| Number of children | None | 179 (22.3%) |
| | 1 | 243 (30.3%) |
| | 2–4 | 318 (39.7%) |
| | 5 or more | 62 (7.7%) |
| Ethnicity | Tigrian | 782 (97.5%) |
| | Others** | 20 (2.5%) |
| Religion | Orthodox | 724 (90.3%) |
| | Muslim | 46 (5.7%) |
| | Others*** | 42 (4.0%) |
| Educational level | Illiterate | 74 (9.2%) |
| | Able to write and read | 107 (13.3%) |
| | Grade 1–8 | 102 (12.7%) |
| | Grade 9–12 | 199 (24.8%) |
| | College and above | 394 (49.1%) |
| Occupational status | Merchant | 224 (27.9%) |
| | Governmental employee | 289 (36.0%) |
| | Non-governmental employee | 84 (10.5%) |
| | Daily labor | 120 (15.0%) |
| | Others**** | 82 (10.6%) |
| Income | Below the poverty line | 164 (20.4%) |
| | Above the poverty line | 638 (79.6%) |
| Psychosocial support | Poor | 356 (44.4%) |
| | Moderate | 308 (38.4%) |
| | Strong | 138 (17.2%) |

**Note:** *Separated, Widowed,

**Amara, Oromo,

***Catholic, Protestant,

****Student, Unemployed.

46.9% currently consume alcohol, 6% currently smoke cigarettes, 5.9% now chew khat, and 50.6% did not meet the recommended physical activity (Table 2).

## Quality life of the study participants

The mean score of the WHOQOL-BREF scale for this study was 92.4, with a Standard Deviation of ±8.5. Of the 802 study participants, 42.4%had poor quality of life.

## Prevalence of erectile dysfunction

The mean score for the International Erectile Function Test (ILEF-5) was 20.9 ± 4.7 (range: 3–25). In this study, the prevalence of ED was 25.4%(95% CI: (22.4, 28.3%)), of which 1.5% had severe ED, 7.3% had moderate ED, and 16.5% had mild ED.

## Factors associated with erectile dysfunction

After bivariate logistic regression analysis, age, income, physical inactivity, current tobacco use, depression, quality of life, diabetes mellitus, hypertension, and neurological disorders met the requirement ($p \leq 0.2$) for multivariable binary logistic regression analysis. Multivariable binary logistic regression analyses show that there has been an association between ED and age, physical inactivity, poor quality of life, and depression.

Respondents whose age was 40 years and above were 10.6 times more likely to develop ED as compared to those under 40 years of age [AOR = 10.74, 95% CI: (7.07, 16.35)]. On the other hand, the odds of developing ED among those physically inactive were 5.9 times higher than those who were physically active [AOR = 3.62, 95% CI: (2.40, 5.45)]. Moreover, the odds of developing ED among study participants who had depression were 3.9 times greater than those who had not depression [AOR = 4.01, 95% CI: (2.22, 7.21)]. Furthermore, the odds of developing ED was 1.6 more likely to occur among those with poor quality of life than those with good quality of life [AOR = 1.59, 95% CI: (1.07, 2.36)] (Table 3).

**Table 2. Distribution of study participants by clinical and lifestyle factors in Adigrat Town, North Ethiopia, 2019 (n = 802).**

| Variables | Frequency or Percentage |
| --- | --- |
| Heart disease | 15 (1.9%) |
| Kidney disease | 21 (2.6%) |
| Diabetic Mellitus | 41(5.1%) |
| Liver disease | 11 (1.4%) |
| Hypertension | 48 (6.0%) |
| Respiratory disease | 21 (2.6%) |
| Neurological disorder | 19 (2.4%) |
| Lower urinary tract symptoms | 14 (1.7%) |
| Depression | 78 (9.7%) |
| Physical inactivity | 406 (50.6%) |
| Ever use of khat | 179 (22.3%) |
| Current use of khat | 47 (5.9%) |
| Ever use of alcohol | 602 (75.1%) |
| Current use of alcohol | 376 (53.1%) |
| Ever use of tobacco | 87 (10.8%) |
| Current use of tobacco | 48 (6.0%) |

**Table 3. Binary logistic regression analysis of factors associated with erectile dysfunction among the male population in Adigrat town, North Ethiopia, 2020 (n = 802).**

| Variables | Category | ED | | OR with 95% CI | |
|---|---|---|---|---|---|
| | | Yes | No | Crude | Adjusted |
| Age | <40 | 87 | 526 | 1 | |
| | ≥40 | 117 | 72 | 9.83 (6.78, 14.24) | 10.74 (7.07, 16.35)* |
| Income | Below poverty line | 35 | 169 | 1.33 (0.5, 1.14) | 0.73 (0.45, 1.20) |
| | Above the poverty line | 129 | 469 | 1 | 1 |
| Physical activity | Ye | 57 | 339 | 1 | 1 |
| | No | 147 | 259 | 3.38 (2.39, 4.77) | 3.62 (2.40, 5.45)* |
| Current tobacco use | Yes | 17 | 31 | 1.66 (0.9, 3.07) | 2.12 (0.97, 4.65) |
| | No | 187 | 567 | 1 | 1 |
| Hypertension | Yes | 20 | 28 | 2.21 (1.22, 4.02) | 0.87 (0.37, 2.04) |
| | No | 184 | 570 | 1 | 1 |
| Diabetic mellitus | Yes | 14 | 27 | 1.56 (0.80, 3.03) | 0.83 (0.33, 2.09) |
| | No | 190 | 571 | 1 | 1 |
| Neurologic disorder | Yes | 9 | 10 | 2.71 (1.09, 6.78) | 1.85 (0.54, 6.39) |
| | No | 195 | 588 | 1 | 1 |
| Depression | Yes | 47 | 31 | 5.48 (3.37, 8.91) | 4.01 (2.22, 7.21)* |
| | No | 157 | 567 | 1 | 1 |
| Quality of life | Poor | 115 | 225 | 2.14 (1.55, 2.96) | 1.59 (1.07, 2.36)* |
| | Good | 89 | 373 | 1 | 1 |

**Note:** 1.00 remained for reference category,

*significance level at p-value <0.05

**Abbreviations:** ED, erectile dysfunction; OR, odds ratio; CI, confidence interval.

## Discussion

The prevalence of ED in the current study was 25.4% (95% CI: (22.4, 28.3%)). The prevalence of erectile dysfunction ranging from 18% to 70% has been recorded worldwide. This finding was lower as compared to studies in Brazil (46.2%) [13], Nigeria (58.9%) [19], (46.9%) [29], New York City (46.3%) [11], Vietnam (66.9%) [16] and Turkey (69.2%) [17], Morocco (54%) [39]. The difference could be attributed to socio-cultural and environmental factors. Also, due to the fact, the age of the study participants is low; the exact prevalence of ED prevalence may be underestimated.

However, the prevalence in the current study was higher compared to studies in the USA (18.4%) [10], Spain (18.9%) [12], and Iran (18.8%) [18]. This variation could be due to the use of different tools in other studies, or it may be a real difference. However, our finding was nearly similar to the finding in Western Australia (25.1%) [14], China (28.3%) [15], Tanzania (24%) [20], and Egypt (23.5%) [21].

This study showed that age was significantly associated with ED. Participants aged 40 years and above were 10.8 times more likely to experience ED than those younger than 40 years of age. This is compatible with the findings of studies conducted in Tanzania [20], Nigeria [29], Egypt [21], and China [15]. This could be because age-related physiological changes in the testicles and the decline in male sex hormones have been attributed to an increased ED incidence in older men [40].

Consistent with studies in the USA [10], Brazil [25], and Singapore [31], our study found that participants who did not meet the recommended weekly physical activity were 5.95 times more likely to have ED than those who did meet the recommended weekly physical activity.

This could be explained by physical exercise that increases nitric oxide bioavailability [28], which is necessary to stimulate smooth muscle relaxation and increase blood flow required for erection [41].

Moreover, this study found that ED was significantly associated with depression. The odds of those participants with depressive symptoms were 3.8 times more likely to have ED than those without depressive symptoms. This result is supported by various studies showing a relationship between ED and depression [13, 17, 23, 24, 26, 30]. Perhaps depressed people may have suffered from ED, as they appear to be highly critical of themselves and have a low degree of sexual appetite because of their feeling of low self-esteem and guilt feeling [42]. This indicates that the cause of ED is primarily due to psychological causes rather than other causes [43]. However, since the study is cross-sectional, it is difficult to ascertain the direction of causality.

In addition, QoL was significantly associated with ED. Those labeled as having poor QoL were 1.6 times more likely to report ED than those labeled as having good QoL. This discovery is reinforced by studies conducted in Vietnam [16] and Egypt [21]. QoL is decreased in men with ED Individuals; because the inability to function sexually can undermine an individual's sense of self-esteem and lead to emotional and marital tension [44]. Medical treatments for ED should be focused on improving the sexuality of individuals with ED [45] by promoting the quality of the patient's life [46].

## Limitation

This study has certain limitations that should be taken into account when interpreting the result. First, since the study design is cross-sectional, this study does not confirm the relationship between cause and effect. Second, even though ED is strongly associated with social stigmas, but the information was collected through the interviewer-administered questionnaire. Because of this, study participants likely underreported ED. Third, we collect data on chronic medical conditions by asking if they had been diagnosed before from any private and public health institutions. Furthermore, the study may be prone to social desirability bias due to sensitive questions related to substance use. Finally, certain potentially significant variables were not included in our study due to omission (e.g., dyslipidemia, drugs used for the treatment of depression, and drugs used for the treatment of cardiovascular diseases).

## Conclusions

The overall prevalence of ED in the study population was very high. Many participants had mild erectile dysfunction. ED was significantly associated with old age, depression, physically inactive, and poor quality of life. It is a known fact that sufferers often do not seek medical help because of the stigma associated with ED but usually suffer in silence. The first step that should encourage men to seek diagnosis and treatment is to overcome any embarrassment or stigma that men and their wives feel. Therefore, it is recommended that the ED diagnosis and treatment service be incorporated into mental health clinics, particularly in the developing world. Health programs in developing nations should be designed to educate and empower individuals on healthy eating, physical activity, and health-seeking behavior.

## Supporting information

**S1 File. Questionnaire.**
(DOCX)

**S2 File. Data set of the prevalence and associated factors of erectile dysfunction using a community based cross-sectional study.**
(SAV)

## Acknowledgments

The authors would like to thank facilitators, data collectors, and the study participants for their dedicated cooperation and made the study possible.

## Author Contributions

**Conceptualization:** Haftom Tesfay Gebremedhin, Hagos Mehari Mezgebo, Gessessew Teklebrhan Geberhiwot, Tesfay Tsegay Gebru, Mulu Ftwi Baraki, Guesh Teklu woledemariam, Tsegu Hailu Gebru, Haileslassie Tesfay Tadese, Gebreslassie Gebreegziabhier Kindeya, Telake Azale.

**Data curation:** Haftom Tesfay Gebremedhin, Hagos Mehari Mezgebo, Gessessew Teklebrhan Geberhiwot, Tesfay Tsegay Gebru, Yowhans Ashebir Tesfamichael, Hailu Belay Ygzaw, Mulu Ftwi Baraki, Guesh Teklu woledemariam, Tsegu Hailu Gebru, Haileslassie Tesfay Tadese, Gebreslassie Gebreegziabhier Kindeya, Telake Azale.

**Formal analysis:** Haftom Tesfay Gebremedhin, Hagos Mehari Mezgebo, Gessessew Teklebrhan Geberhiwot, Tesfay Tsegay Gebru, Yowhans Ashebir Tesfamichael, Hailu Belay Ygzaw, Mulu Ftwi Baraki, Guesh Teklu woledemariam, Tsegu Hailu Gebru, Haileslassie Tesfay Tadese, Gebreslassie Gebreegziabhier Kindeya, Telake Azale.

**Investigation:** Haftom Tesfay Gebremedhin.

**Methodology:** Haftom Tesfay Gebremedhin, Hagos Mehari Mezgebo, Gessessew Teklebrhan Geberhiwot, Tesfay Tsegay Gebru, Yowhans Ashebir Tesfamichael, Hailu Belay Ygzaw, Mulu Ftwi Baraki, Guesh Teklu woledemariam, Tsegu Hailu Gebru, Haileslassie Tesfay Tadese, Gebreslassie Gebreegziabhier Kindeya, Telake Azale.

**Project administration:** Haftom Tesfay Gebremedhin.

**Resources:** Haftom Tesfay Gebremedhin.

**Software:** Haftom Tesfay Gebremedhin, Hagos Mehari Mezgebo, Gessessew Teklebrhan Geberhiwot, Tesfay Tsegay Gebru, Yowhans Ashebir Tesfamichael, Hailu Belay Ygzaw, Mulu Ftwi Baraki, Guesh Teklu woledemariam, Tsegu Hailu Gebru, Haileslassie Tesfay Tadese, Gebreslassie Gebreegziabhier Kindeya, Telake Azale.

**Supervision:** Haftom Tesfay Gebremedhin.

**Visualization:** Haftom Tesfay Gebremedhin.

**Writing – original draft:** Haftom Tesfay Gebremedhin, Hagos Mehari Mezgebo, Gessessew Teklebrhan Geberhiwot, Tesfay Tsegay Gebru, Yowhans Ashebir Tesfamichael, Hailu Belay Ygzaw, Mulu Ftwi Baraki, Guesh Teklu woledemariam, Tsegu Hailu Gebru, Haileslassie Tesfay Tadese, Gebreslassie Gebreegziabhier Kindeya, Telake Azale.

**Writing – review & editing:** Haftom Tesfay Gebremedhin, Hagos Mehari Mezgebo, Gessessew Teklebrhan Geberhiwot, Tesfay Tsegay Gebru, Yowhans Ashebir Tesfamichael, Hailu Belay Ygzaw, Mulu Ftwi Baraki, Guesh Teklu woledemariam, Tsegu Hailu Gebru, Haileslassie Tesfay Tadese, Gebreslassie Gebreegziabhier Kindeya, Telake Azale.

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
