## [Decision Letter · Decision Letter 0]

29 Sep 2020

PONE-D-20-24723

Erectile dysfunction and its associated factors among the male population in Adigrat Town, Tigrai Region, Ethiopia: A cross-sectional study

PLOS ONE

Dear Dr. Gebremedhin,

Thank you for submitting your manuscript to PLOS ONE. After careful consideration, we feel that it has merit but does not fully meet PLOS ONE’s publication criteria as it currently stands. Therefore, we invite you to submit a revised version of the manuscript that addresses the points raised during the review process.

In particular, both reviewers make the point that ED prevalence is rather low, which is likely due to the age of the study participants and the inclusion criteria. This needs further discussion.

We look forward to receiving your revised manuscript.

Kind regards,

Sabine Rohrmann

Academic Editor

PLOS ONE

Journal Requirements:

2. Please include a copy of Table 4 which you refer to in your text on page 15. (Table 1 2x)

Reviewers' comments:

Reviewer's Responses to Questions

**Comments to the Author**

1. Is the manuscript technically sound, and do the data support the conclusions?

Reviewer #1: Yes

Reviewer #2: Yes

2. Has the statistical analysis been performed appropriately and rigorously? 

Reviewer #1: Yes

Reviewer #2: Yes

3. Have the authors made all data underlying the findings in their manuscript fully available?

Reviewer #1: Yes

Reviewer #2: Yes

4. Is the manuscript presented in an intelligible fashion and written in standard English?

Reviewer #1: Yes

Reviewer #2: Yes

5. Review Comments to the Author

Reviewer #1: This is an interesting study assessing the prevalence of ED in an Ethiopian population. The authors found a prevalence of ED of approximately 25%. Furthermore, the authors found that ED is associated with depression, arterial hypertension and physical inactivity, findings that are in line with the literature.

Some points need to be addressed

1. The prevalence of ED is rather low. This theoretically is attributed to the generally low age of study participants. Indeed this rate is in line with studies assessing the ED prevalence in younger populations. A comment should be made in the discussion section stating that the age of the study participants is rather low and thus the true ED prevalence may be underestimated.

2. Dyslipidemia is a common risk factor for ED. Are there any data regarding the prevalence of dyslipidemia in the study participants? And if yes, was there any association with ED prevalence and severity?

3. Depression is commonly associated with increased risk for ED. Drugs for the treatment of depression are often accompanied by ED as a side effect. Was there any relevant association?

4. Several drugs for the treatment of cardiovascular diseases and risk factors (such as beta-blockers and diuretics) are related with increased risk for ED. If fdata are available, please provide any relevant information regarding the use of such drugs and potential relation with ED prevalence.

5. The manuscript is in need of grammatical revision. Please check.

Reviewer #2: Important is the study by Gebremedhin et. al., which focuses on an often neglected health problem. Often the focus is on diseases that are life-threatening in low-income countries, while the morbidity rate attributable to other non-life-threatening problems is also too high.

- Include the confidence interval of the prevalence in the abstract.

- Census 2007 is too old for a rapidly growing population. Please use predicted population.

- " Who had been engaged in sexual activity in a preceding month " is a defining term for the inclusion of your study participants. This is a strong term that may have excluded many people with erectile dysfunction (ED), which would have increased the prevalence if they had been included. Do not believe that your findings could show a severely underestimated prevalence.

- While your target study participants were individuals, you targeted at a household. This looks a bit superfluous and unnecessary. Other options will have been sought.

- Why was "bivariate regression"? That somehow looks like a tradition. This is necessary if you have a smaller sample but more variables and want to select fewer variables. This is to counteract unstable estimates when you fit more variables into a small sample. However, you have a sufficient sample size for the number of variables. This tradition is also problematic. For example, a variable may not be significant if you perform bivariate regression, but it could have an important interaction effect if you use it with other variables. Do not follow this tradition.

- Minor: I think you can reduce the content of your tables (e.g. in table 2 you can have only one column and one row for each variable. The answers 'Y/N' are not necessary. And you also make the frequency and percentage in one column)

- Quality of life pie chart is not necessary

6. PLOS authors have the option to publish the peer review history of their article (what does this mean?). If published, this will include your full peer review and any attached files.

Reviewer #1: **Yes: **Konstantinos Imprialos

Reviewer #2: No

---

## [Author Response · Author response to Decision Letter 0]

8 Oct 2020

Response: Modified according to the PLOS ONE's style requirements.

2. Please include a copy of Table 4 which you refer to in your text on page 15. (Table 1 2x)

 Response: It was a slip of pen that we wrote “Table 4” in both the reference and the title of the table and we have corrected the mistake and revised it to “Table 3” (page 15, line 257-259).

3. The prevalence of ED is rather low. This theoretically is attributed to the generally low age of study participants. Indeed this rate is in line with studies assessing the ED prevalence in younger populations. A comment should be made in the discussion section stating that the age of the study participants is rather low and thus the true ED prevalence may be underestimated. 

Response: We agree with this comment. Accordingly, we have stated it in discussion part (page 18, line 281-283).

4. Dyslipidemia is a common risk factor for ED. Are there any data regarding the prevalence of dyslipidemia in the study participants? And if yes, was there any association with ED prevalence and severity? 

Response: We accepted the comment and have included it as the limitation of the study in the limitation section of the manuscript (page 19, line 324-326).

5. Depression is commonly associated with increased risk for ED. Drugs for the treatment of depression are often accompanied by ED as a side effect. Was there any relevant association? 

Response: Thank you for noticing this. In fact medications used to treat depression have an effect on an individual’s sexual performance, but we omitted them. We have therefore included them as a limitation of the study (page 19, line 324-326).

6. Several drugs for the treatment of cardiovascular diseases and risk factors (such as beta-blockers and diuretics) are related with increased risk for ED. If data are available, please provide any relevant information regarding the use of such drugs and potential relation with ED prevalence. 

Response: It is true that medications used to treat cardiovascular disease are important factors for ED, and we have included them in the limitation part of the study (page 19, line 324-326).

7. The manuscript is in need of grammatical revision. Please check. 

Response: Checked.

8. Include the confidence interval of the prevalence in the abstract. 

Response: We agree with this comment. Accordingly, we have included the confidence interval of the prevalence in the abstract part (page3, line 53).

9. Census 2007 is too old for a rapidly growing population. Please use predicted population.

 Response: Thank you for pointing out this and we have updated this with the latest (2017) estimated population of the Town from the report of the Central Statistical Agency of the Federal Democratic Republic of Ethiopia (page 6, line 113-115).

10. " Who had been engaged in sexual activity in a preceding month " is a defining term for the inclusion of your study participants. This is a strong term that may have excluded many people with erectile dysfunction (ED), which would have increased the prevalence if they had been included. Do not believe that your findings could show a severely underestimated prevalence.

 Response: The instrument (The International Index of Erectile Function Questionnaire (IIEF-5)) used to determine erectile dysfunction measures the erectile function over the last month and that is why we used sexual activity in the last moth preceding the data collection as inclusion criteria. 

11. While your target study participants were individuals, you targeted at a household. This looks a bit superfluous and unnecessary. Other options will have been sought.

 Response: Thank you for the suggestion. It was hard to get a list of male individuals in the Town. Instead of selecting individuals from the sampling frame, we used systematic randomized sampling to select study unit (households) and simple randomized sampling to select study participants if we found more than one eligible study participants. In this method, sampling error might be unavoidable, thus minimizing the error that we used the design to maximize sample size. 

12. Why was "bivariate regression"? That somehow looks like a tradition. This is necessary if you have a smaller sample but more variables and want to select fewer variables. This is to counteract unstable estimates when you fit more variables into a small sample. However, you have a sufficient sample size for the number of variables. This tradition is also problematic. For example, a variable may not be significant if you perform bivariate regression, but it could have an important interaction effect if you use it with other variables. Do not follow this tradition. 

Response: We appreciate your suggestion. We will consider it for the future.

13. Minor: I think you can reduce the content of your tables (e.g. in table 2 you can have only one column and one row for each variable. The answers 'Y/N' are not necessary. And you also make the frequency and percentage in one column) Response: Great point. We have modified our tables accordingly.

14. Quality of life pie chart is not necessary. 

Response: Agreed. We removed it.

---

## [Decision Letter · Decision Letter 1]

2 Nov 2020

Erectile dysfunction and its associated factors among the male population in Adigrat Town, Tigrai Region, Ethiopia: A cross-sectional study

PONE-D-20-24723R1

Dear Dr. Gebremedhin,

We’re pleased to inform you that your manuscript has been judged scientifically suitable for publication and will be formally accepted for publication once it meets all outstanding technical requirements.

Kind regards,

Sabine Rohrmann

Academic Editor

PLOS ONE

Additional Editor Comments (optional):

Reviewers' comments:

Reviewer's Responses to Questions

**Comments to the Author**

1. If the authors have adequately addressed your comments raised in a previous round of review and you feel that this manuscript is now acceptable for publication, you may indicate that here to bypass the “Comments to the Author” section, enter your conflict of interest statement in the “Confidential to Editor” section, and submit your "Accept" recommendation.

Reviewer #1: (No Response)

2. Is the manuscript technically sound, and do the data support the conclusions?

Reviewer #1: Yes

3. Has the statistical analysis been performed appropriately and rigorously? 

Reviewer #1: Yes

4. Have the authors made all data underlying the findings in their manuscript fully available?

Reviewer #1: Yes

5. Is the manuscript presented in an intelligible fashion and written in standard English?

Reviewer #1: Yes

6. Review Comments to the Author

Reviewer #1: (No Response)

7. PLOS authors have the option to publish the peer review history of their article (what does this mean?). If published, this will include your full peer review and any attached files.

Reviewer #1: **Yes: **Konstantinos Imprialos

---

## [Editor Report · Acceptance letter]

10 Mar 2021

PONE-D-20-24723R1 

Erectile dysfunction and its associated factors among the male population in Adigrat Town, Tigrai Region, Ethiopia: A cross-sectional study 

Dear Dr. Gebremedhin:

I'm pleased to inform you that your manuscript has been deemed suitable for publication in PLOS ONE. Congratulations! Your manuscript is now with our production department. 

Kind regards, 

on behalf of

Dr. Sabine Rohrmann 

Academic Editor

PLOS ONE